# Curcumin Sensitises Cancerous Kidney Cells to TRAIL Induced Apoptosis via Let-7C Mediated Deregulation of Cell Cycle Proteins and Cellular Metabolism

**DOI:** 10.3390/ijms23179569

**Published:** 2022-08-24

**Authors:** Ismael Obaidi, Alfonso Blanco Fernández, Tara McMorrow

**Affiliations:** 1NatPro Centre for Natural Product Research, School of Pharmacy and Pharmaceutical Sciences, Trinity College Dublin, D02 W272 Dublin, Ireland; 2College of Pharmacy, University of Babylon, Babylon 51002, Iraq; 3Flow Cytometry Core Technology, Conway Institute of Biomolecular and Biomedical Research, University College Dublin, D04 V1W8 Dublin, Ireland; 4Centre for Toxicology, School of Biomedical and Biomolecular Sciences, Conway Institute, University College Dublin, D04 V1W8 Dublin, Ireland

**Keywords:** curcumin, apoptosis, chemosensitisation, kidney cancer, TRAIL, let-7C, cell cycle, cellular metabolism

## Abstract

Targeted therapies are the most attractive options in the treatment of different tumours, including kidney cancers. Such therapies have entered a golden era due to advancements in research, breakthroughs in scientific knowledge, and a better understanding of cancer therapy mechanisms, which significantly improve the survival rates and life expectancy of patients. The use of tumour necrosis factor (TNF)-related apoptosis inducing ligand (TRAIL) as an anticancer therapy has attracted the attention of the scientific community and created great excitement due to its selectivity in targeting cancerous cells with no toxic impacts on normal tissues. However, clinical studies disappointingly showed the emergence of resistance against TRAIL. This study aimed to employ curcumin to sensitise TRAIL-resistant kidney cancerous ACHN cells, as well as to gain insight into the molecular mechanisms of TRAIL sensitization. Curcumin deregulated the expression of apoptosis-regulating micro Ribonucleic Acid (miRNAs), most notably, let-7C. Transfecting ACHN cells with a let-7C antagomir significantly increased the expression of several cell cycle protein, namely beta (β)-catenin, cyclin dependent kinase (CDK)1/2/4/6 and cyclin B/D. Further, it overexpressed the expression of the two key glycolysis regulating proteins including hypoxia-inducible factor 1-alpha (HIF-1α) and pyruvate dehydrogenase kinase 1 (PDK1). Curcumin also suppressed the expression of the overexpressed proteins when added to the antagomir transfected cells. Overall, curcumin targeted ACHN cell cycle and cellular metabolism by promoting the differential expression of let-7C. To the best of our knowledge, this is the first study to mechanistically report the cancer chemosensitisation potential of curcumin in kidney cancer cells via induction of let-7C.

## 1. Introduction

Kidney cancer remains one of the top ten cancers worldwide with the current estimated deaths of 13,920 in the United States [1]. Despite the advancement in cancer treatments, including in renal cancers, resistance is still problematic as it leads to poor prognosis, with only a 5% overall five-year survival rate in advanced renal cell carcinoma (RCC) [2].

The use of tumour necrosis factor (TNF)-related apoptosis inducing ligand (TRAIL) as an anticancer therapy has attracted the attention of the scientific community and created great excitement because it selectively targets cancer cells with minimal or no effects on noncancerous counterparts [3,4]. TRAIL is a cytokine belonging to the TNF superfamily that mainly activates the extrinsic pathway of apoptosis upon binding to its cognate receptors, known as death receptors (DRs) including DR4 and DR5 [5]. Upon the binding to DR4 or DR5, TRAIL recruits Fas Associated Death Domain (FADD) at the cytoplasmic death domain (DD) of the receptors, causing activation of the initiator caspases, such as caspase 8 and caspase 10. The complex of DRs with FADD and caspase 8/10 forms what’s called Death Inducing Signaling Cascade (DISC) which is the trigger of apoptotic events that ends with activation of the executioner caspases 3/7. Several proteases and nucleases are then activated, causing protein destruction, nuclear fragmentation, collapse of cytoplasmic membrane, and formation of apoptotic bodies. In some instances, the activated caspase 8 truncates the mitochondrial B-cell lymphoma 2 (Bcl-2)-related proapoptotic protein (Bid) triggering the intrinsic or the mitochondrial pathway of apoptosis [4,6,7]. Interestingly, normal non-transformed cells resist TRAIL’s proapoptotic action via different resistance mechanisms. Firstly, they overexpress TRAIL-R3 decoy receptors while downregulate apoptosis-mediated DR4/5. Secondly, they upregulate several anti-apoptotic proteins including Inhibitors of Apoptosis Proteins (IAP) [8], cellular FLICE-like inhibitory protein (cFLIP) and Bcl-2 [9]. Downregulation of one of these antiapoptotic proteins is not sufficient to sensitize non-transformed cells to apoptosis, reflecting the multiple mechanisms involved in TRAIL resistance in normal cells. In contrast, most cancer cells overexpress one of anti-apoptotic proteins leading to resistance development. Perturbation of one of these mechanisms is sufficient to overcome TRAIL resistance and sensitise cancer cells to TRAIL-induced apoptosis. Taken together, the abundance of TRAIL resistance mechanisms in normal non-transformed cells can provide a wide safety margin for TRAIL-based combination therapy which targets one anti-apoptotic pathway in cancerous cells [9].

Disappointingly clinical trials have shown that, as a monotherapy, TRAIL is not an effective anticancer agent against multiple types of malignancies due to the emergence of resistance by tumour cells. Thus, there is an unmet need to restore the sensitivity of cancer cells to TRAIL [10,11]. Much research has been conducted to achieve TRAIL sensitization using combinatorial regimens of TRAIL with various compounds including natural products [12,13,14,15]. One of the successful approaches involved targeting cancer cells with a combinatorial regimen of curcumin and TRAIL [16,17].

Curcumin is a polyphenolic yellow spice derived from the rhizomes of *Curcuma longa* L. plant [18]. It elicits a myriad of therapeutic benefits [19,20,21,22], including chemoprevention, anticancer, and chemosensitisation [23,24,25]. Structurally, curcumin or diferuloylmethane-[1,7-bis(4-hydroxy-3-methoxyphenyl)-1,6-hepadiene-3,5-dione] shows the presence of bis-α, β-unsaturated β-diketone, which provides electron scavenging and antioxidant properties by losing an electron. Curcumin detoxifies free radicals such as superoxide, hydroxyl, peroxyl radicals via hydrogen abstraction or electron donation from its three active sites. This leads to formation of phenoxyl radicals which exist as keto-enol tautomeric mixtures. Curcumin is regenerated by hydrogen donners such as water-soluble vitamin C [26] (Figure 1).

Curcumin can target cancer cells due to its direct effect on cell death machinery, cell cycle proteins and survival signaling pathways such as phosphatidylinositol 3-kinase/protein kinase B (PI3K/Akt), nuclear factor kappa light chain enhancer of activated B cells (NF-κB), mitogen-activated protein kinases (MAPK), Wingless-related integration site (Wnt)-Beta catenin (β-catenin), and Signal Transducer and Activator of Transcription-3 (STAT-3) [27]. Curcumin also induces cell death via caspase dependent and independent mechanisms [28,29]. It can induce apoptosis at low concentrations by down regulation of the proteasome system and increasing the level of reactive oxygen species (ROS). Whereas, it promotes necrotic cell death at high concentrations by curtailing ROS and depletion of the adenosine triphosphate (ATP) levels [30]. Emerging epidemiological, animal, and clinical studies show that consumption of curcumin can reduce the risk of some types of cancers [25,31,32], while a large body of evidence has unraveled the anticancer mechanisms of curcumin [2,31,33,34,35] including chemosensitisation of cancer cells to TRAIL’s proapoptotic action [16,17,36,37,38]. Curcumin can epigenetically modulate gene expression of cancer cells by targeting DNA methylation, histone acetylation, or micro ribonucleic acid (miRNAs) machineries. There is a significant interplay between curcumin and miRNAs expression. For instance, stable expression of miR-378 synergistically promoted the response of glioblastoma multiforme to curcumin [39], whereas targeting lung cancer cells with curcumin substantially downregulated the expression of the oncogenic miRNA-21 while upregulating the oncosuppressive miRNA 192 and 215 [40]. Further, miRNA can be a potential biomarker to assess the therapeutic benefits of curcumin against bladder cancer cells [41].

miRNAs are a class of a highly conserved small (19–25 nucleotides) non protein-coding RNA [42]. They regulate about 60% of total human genes [43], and can silence gene expression by induction of degradation or translational inhibition of messenger RNA (mRNA). miRNAs regulate a plethora of biological processes such as proliferation, stress resistance, metabolism, and cell death. They have pro- and anti-apoptotic activities [43]. Let-7C is a member of an evolutionarily conserved family of 13 homologous miRNAs encoded by the *let-7* gene, which is mutated in most human cancers [44]. Let-7C can be employed as a prognostic factor as well as a treatment option for RCC [45]. For instance, a clinical study by Heinzelmann et al. showed that let-7C expression was significantly reduced in highly aggressive metastatic clear cell RCC (ccRCC) compared to the non-metastatic cases. Thus, let-7C can be utilized as a prediction tool for disease progression and survival metrics including progression-free and overall survival rates [46]. Another study by Peng et al. reported that the expression of the oncosuppressive let-7B and let-7C were low in surgically resected RCC specimens resistant to the chemotherapeutic agent 5-fluorouracil (5-FU). The overexpression of these miRNAs significantly sensitised the RCC cells to 5-FU by targeting the RAC-beta serine/threonine-protein kinase (*AKT2*) gene, and thus confirming the chemosensitising potential of let-7B/C [47].

This study was aimed at investigating the curcumin chemosensitising mechanisms to TRAIL by examining the involvement of miRNAs. We also aimed to link curcumin-induced miRNA deregulation and modulation of vital cellular processes including cell cycle and metabolism. Therefore, we believe that this study represents a logical and important advance to our previous study [16], in which we reported that curcumin sensitised the renal cancerous ACHN to TRAIL’s proapoptotic effect via the activation of ROS/DR4 and ROS/Jun N-terminal kinase (JNK)/C/EBP homologous protein (CHOP) pathways.

## 2. Results

### 2.1. Curcumin or Curcumin/TRAIL Combination Induced Irreversible ACHN Cell Death

In our previous study [16], we reported that curcumin at 25 µM sensitised ACHN cells, but not epithelial RPTEC/TERT1 cells, to apoptosis following the combination treatment with 50 ng/mL TRAIL. In the present study, we treated the cells as indicated for 24 h, the culture medium containing treatments was then replaced with fresh medium for five days followed by cell fixation, staining with crystal violet, photographing and absorbance measurements. We found that curcumin, by itself or in the combination with TRAIL, induced irreversible cell death rather than just growth inhibition, with no cellular recovery even after removal of treatments. Clearly, wells treated with curcumin or curcumin + TRAIL showed a colourless appearance indicating no cellular growth. Whereas the purple staining refers to the abundance of healthy cells in the wells treated with dimethylsulfoxide (DMSO) or TRAIL alone (Figure 2a). The color intensity was then measured spectrophotometrically using a plate reader and data showed the relative cellular growth compared to the untreated control (Figure 2b).

### 2.2. Curcumin Induced ACHN Cell Cycle Arrest

Curcumin has been documented to induce cell cycle arrest in many cancer cells via different mechanisms [48,49,50,51,52]. In the present study, we investigated the chemosensitising potential of curcumin on the TRAIL resistant ACHN cells by examining its impact on cell cycle progression, thus we undertook cell cycle analysis using flow cytometry by measuring DNA content. We found that targeting the cancerous ACHN cells with curcumin alone or in a combination with TRAIL markedly inhibited ACHN cell cycle progression (Figure 3a,b). The effect of curcumin can be observed as early as eight hours post treatment when the cells accumulated at quiescence/growth 1 (G0/G1)-phase while less cells were observed at synthesis (S)-phase compared to DMSO or TRAIL only treated cells. At 12 h treatment, the ACHN cell cycle was significantly halted at G0/G1 thus less cells can be observed at growth 2/mitosis (G2/M) stage following curcumin or curcumin + TRAIL treatment. Curcumin or curcumin + TRAIL treated cells were shown to accumulate at S and G2/M phases at 24 h post treatment compared to TRAIL or the untreated control. Notably, there are no differences in the distribution of cells among all cell cycle phases between curcumin only or curcumin + TRAIL combination treatments (Figure 3a,b).

### 2.3. Curcumin Promoted Mitochondrial Activity, but Not Mitochondrial Mass

We measured the effect of the indicated treatments on both mitochondrial activity and mass using MitoTracker Red CMXRos and MitoTracker Green (Thermo Fischer Scientific, Waltham, MA, USA), respectively by Flow cytometry. Treating ACHN cells with either curcumin or the curcumin + TRAIL combination clearly increased ACHN mitochondrial activity compared to TRAIL alone or untreated control (Figure 4a). The combination of TRAIL with curcumin, but not curcumin only treatment, increased the mitochondrial mass (Figure 4b). Curcumin enhanced mitochondrial activity 24 h post treatment, however no changes were detected earlier than 24 h (Figure 4c(i)) which indicates that, unlike its effect on cell cycle, curcumin had a “late” rather than “early” effect on mitochondrial activity. No change in the MitoTracker Green intensity (mitochondrial mass) was detected following treatment with curcumin (Figure 4c(ii)). This indicates that curcumin increased the mitochondrial activity rather than altering mitochondrial mass.

### 2.4. Differential Expression of Apoptosis Regulating miRNAs by Curcumin or Curcumin/TRAIL Combination Treatments

To profile the differential expression of various apoptosis-regulating miRNAs following treatment of ACHN cells with curcumin, TRAIL or curcumin + TRAIL combination, we carried out miRNA microarray analysis using miScript miRNA polymerase chain reaction (PCR) array. In the resulting heat map, the upregulated miRNAs appear in red, the downregulated in green while the non-affected miRNAs appear in black. Samples from control, curcumin and TRAIL treatment tend to cluster together while curcumin+ TRAIL combination samples were slightly divergent. In this array, the effect of curcumin, TRAIL, and curcumin+ TRAIL treatments were investigated on a total of 84 apoptosis-regulating miRNAs. Following normalisation, a twofold change threshold was set for both upregulated and downregulated miRNAs. miRNA 30d was found to be upregulated by TRAIL or curcumin treatments. Let-7C and miRNA 153 were significantly upregulated by curcumin and curcumin + TRAIL combination. miRNA 203a was upregulated following treatment with curcumin + TRAIL combination only. Let-7C was shown to be the most significantly upregulated miRNA by curcumin or curcumin+ TRAIL combination. Interestingly, the oncosuppressive miRNA 542 was significantly downregulated while the oncogenic miRNA 210 was significantly upregulated by TRAIL only but not curcumin or curcumin + TRAIL combination. In fact, miRNA 210 was the most significantly upregulated miRNA following TRAIL treatment, suggesting that TRAIL might activate non-canonical survival pathways which were suppressed by curcumin (Figure 5 and Appendix A).

### 2.5. Validation of Let 7C Differential Expression in ACHN Cells Exposed to TRAIL, Curcumin or Curcumin + TRAIL

We carried out validation experiment of the miRNA array data using single assay TaqMan-based real-time quantitative reverse transcription (qRT)-PCR analysis (Figure 6). We selected let-7C for validation analysis as it was identified as having the greatest expression change. We found that curcumin, alone or in a combination with TRAIL significantly upregulated the let-7C miRNA compared to TRAIL or untreated negative control, thus confirming the results of our microarray data. However, no significant difference was observed between curcumin and curcumin + TRAIL treatments, suggesting that curcumin stimulates let-7C expression (Figure 6).

### 2.6. In Silico miRNA-Targeted Gene Prediction Analysis

miRNAs play an integral role in regulating gene expression via binding if the 3′ untranslated region of mRNA, thus inducing mRNA degradation and post-transcriptional inhibition. Further analysis showed that the top five significantly upregulated miRNAs by curcumin were let-7C, miRNA 3b, miRNA 30d, miRNA 31, and miRNA 153. Their potential pathway targets were predicted using online algorithms such as mirbase, miRDB, PicTar, MirTargetLink2.0 and TargetScan. Putative target mRNAs were selected by applying cut off filters for each database. For instance, target score ≥ 70 was used as a cut off filter in miRDB, aggregate ≥ 0.5 was set in Target Scan, and a score of ≥2 in PicTar. The predicted targets from each database were overlapped and grouped based on their activities. Common target pathways affected by the selected miRNA are cell cycle, cellular metabolism, angiogenesis, and cellular adhesion molecules (Appendix A).

### 2.7. The Expression of a Group of Cell Cycle Proteins Was Increased by the Let-7C Inhibitor while the Expression of the Same Group of Proteins Was Downregulated by Curcumin

To investigate whether curcumin altered the expression of cell cycle proteins via upregulation of let-7C, we transfected ACHN cells with a let-7C antagomir or a negative control miRIDIAN microRNA. qRT-PCR was employed to measure post transfection let-7C gene expression. The optimal time and concentration of let-7C antagomir for transfection were determined to be 100 nM for 24 h (Appendix A). Following this, Western blotting was used to investigate the effect of let-7C inhibition and/or curcumin treatment on cell cycle proteins, known to be regulated by let-7C (Figure 7a). This group of proteins includes β catenin, CDK1, CDK2, CDK4, CDK6, cyclin B, and cyclin D1. Interestingly, the inhibition of let-7C increased the expression of these cell cycle proteins. The combination of curcumin with the let-7C antagomir prevented the upregulation of the cell cycle proteins compared to let-7C antagomir transfection alone (Figure 7a). The effects of these treatments on the Western blot data were further analysed by measuring band intensity densitometrically using the Image J software 1.50i (National Institutes of Health, Bethesda, MD, USA) (http://imagej.nih.gov/ij, accessed on 23 February 2020) (Figure 7b).

### 2.8. The Expression of a Group of Glycolysis-Regulating Proteins Increased by the Let-7C Inhibitor, while the Expression of the Same Group of Proteins Was Suppressed by Curcumin

To determine whether the effect of curcumin on cellular metabolism is linked to the upregulation of let-7C, the cells were transfected with let-7C antagomir or a miRIDIAN microRNA negative control. Western blotting was employed to investigate the effect of let-7C inhibition on the expression of two important glycolysis regulating proteins including hypoxia-inducible factor 1-alpha (HIF-1α) and pyruvate dehydrogenase kinase 1 (PDK1) (Figure 8a). This investigation was undertaken as HIF-1α and PDK1 mRNAs are both known targets of let-7C [53,54]. Interestingly, the inhibition of let-7C increased the expression of both proteins compared to the mock transfection treatment. In the cells treated with curcumin, the let 7C antagomir is unable to increase the expression of the HIF-1α and PDK1 proteins (Figure 8a). The effects of the treatments on the Western blot data were analysed and compared by measuring band intensity densitometrically using the image J software (Figure 8b).

## 3. Discussion

TRAIL is in the spotlight as a promising anticancer agent that specifically targets cancer cells without affecting normal counterparts. However, drug resistance has emerged in many cancerous cells against TRAIL due to its short duration of action and activation of non-canonical survival pathways. Yet, bortzomib and other proteasome inhibitors are clinically available options which counteract resistance to TRAIL. However, as a chemotherapeutic agent, bortezomib causes serious adverse effects [55,56,57], which necessitates pursuing alternatives for TRAIL sensitising activity but with less or no damaging impacts on normal tissues [58,59,60]. Curcumin has attracted the interest of the scientific community due to its diverse therapeutic applications, including anticancer activity with limited side effects. In our previous study, we reported that curcumin has a chemopreventive potential by suppression of DNA mutations, inflammation and tissue dedifferentiation induced by potassium bromate, a genotoxic carcinogen [23]. Further, curcumin was demonstrated to have a chemosensitising property via the induction of ROS, which in turn led to upregulation of DR4 and activation of the JNK/CHOP pathway [16]. Therefore, the main goal of this study was to further investigate the molecular mechanisms of curcumin induced ACHN sensitisation to TRAIL and to explore the role of miRNAs in the chemosensitisation process.

In the present study, we showed that curcumin inhibited the long-term survival and induced cell cycle arrest of the cancerous ACHN cells. The inhibition of clonal outgrowth and induction of cell death were also shown with other cancer chemosensitising agents, such as withanolide E, cyanocycline A, or antibiotic M259 [59,61]. Curcumin also affected the progression of the ACHN cell cycle, which is consistent with previous reports [48,62,63,64,65]. Interestingly the uptake and accumulation of curcumin and, thus toxicity is significantly higher in cancerous than normal healthy cells [66]. A study by Ghosh and Ryan found that upon entering Hep G2 cells, curcumin migrates to the nucleus where it is localized, concentrated, and sequestered by nucleophosmin, a nucleolus resident protein [67]. This leads to the translocation of the cyclin-dependent kinase inhibitor 2A (p14ARF), a nucleophosmin confined protein [67] from the nucleolus to the nucleoplasm where it binds and inactivates murine double minute 2 (MDM2) [68,69]. The sequestration of MDM2 inhibits the ubiquitination and proteasomal degradation of p53, which results in p53 activation and subsequently cell cycle arrest and apoptosis [70]. In our previous study, we found that both p53 and CDK1 were deregulated by curcumin [16].

In addition to the activation of p53, curcumin arrests cell cycle progression of tumour cells by targeting miRNAs [71,72]. A key miRNA mobilised by curcumin is let-7C [43,72,73,74,75]. In the present study, several miRNAs including let-7C were induced by curcumin. Other scientific studies have also shown that curcumin induced the expression of let-7C [72,73,76,77]. The upregulation of let-7C is associated with induction of cell cycle arrest in various tumour cells by regulating the expression of various cell cycle proteins. Let-7-C can affect the expression of β-catenin either by targeting the upstream gene Dishevelled Segment Polarity Protein 3 (*DVL3*) [78], or by negatively regulating the expression of the Wnt signaling pathway, namely Wnt1, T-cell factor (TCF) and cyclin D1 [79,80]. Inhibition of Wnt/β-catenin signaling by let-7C can sensitise tumour cells to chemotherapeutic agents [81]. Other possible cell cycle inhibitory mechanisms are modulation of Endoplasmic Reticulum proteins, and alteration of phospho- extracellular signal-regulated kinase (p-ERK) signaling pathways [82]. Another study reported that let-7C affected cell cycle and growth of hepatocellular carcinoma cells via interference with the differential expression of several cell cycle proteins including M-phase inducer phosphatase 1 (CDC25A), cyclin D1, cyclin dependent kinase (CDK) 6, retinoblastoma (pRb) and E2F transcription factor 2 (E2F2) [83]. Cyclin B2, cyclin E2 and CDK4 were reported to be potential targets of let-7C by inhibition of the PI3K/Akt/forkhead box transcription factors (FOXO) signaling pathway [84]. Similarly, let-7C inhibited the cell cycle at G1/S phase by targeting and modulating p15/p16/CDK4/E2F1 pathway of nasopharyngeal carcinoma cells [85]. Furthermore, the let-7 family targets cell cycle control proteins such as CDK1, CDK 2, CDK6 and cyclin B1 in various tumours including renal cancer cells [86,87]. Let-7C also impacts cell cycle progression at G1 by targeting cyclin D/CDK4 and CDK6, inhibiting G1/S progression by targeting cyclin E/CDK2, and interfering with G2/M by inhibiting cyclin B/CDK1 [88].

Besides the impact on the cell cycle, we found that curcumin increased the mitochondrial activity of ACHN cells. Importantly, targeting mitochondria has been considered as one of the attractive anticancer strategies to halt tumour initiation and progression [89]. Interference with tricarboxylic acid (TCA) cycle, electron transport chain, ROS production, mitochondrial turnover, and restoring mitochondrial-driven apoptosis are the main strategies to harness mitochondria against tumour cells [90]. We previously reported that curcumin induced a redox imbalance by increasing ROS production and activation of the mitochondrial pathway of apoptosis thus sensitising ACHN cells to TRAIL induced death [16]. Similarly, other studies found that curcumin activated the TCA cycle [91,92] while reducing glucose consumption and lactate production. Curcumin can also induce reprogramming of tumour metabolism via targeting the ATP machinery of mitochondria, leading to the activation of apoptosis due to energy shutdown [93]. In this study, we have shown that curcumin targets the key glycolysis regulatory proteins HIF-1α and PDK-1 by the upregulation of let-7C. Our result is consistent with the Ma et al. study which found that let-7C inhibited aerobic glycolysis in cancer cells by targeting PDK1 and HIF-1α thus inhibited Lin-28-induced glycolysis and tumour progression [54]. It is well established that the rate of glycolysis in cancerous cells is higher than normal cells, even in the presence of atmospheric oxygen. This aerobic glycolysis is termed “the Warburg effect” [94], whereby the transcriptional factor HIF-1α plays an integral role in activating several glycolysis induction pathways including Myc and PI3K/AKT/mTOR [95,96,97]. The upstream activators of HIF-1α, such as PI3K/Akt, ERK, and adenosine 5′-monophosphate- (AMP-) activated protein kinase (APK), increase the cancer cell glycolysis rate, while p53 and Von Hippel-Lindau syndrome (VHL) block HIF-1α induced glycolysis in cancer cells. HIF-1α induces the expression of the glycolysis-regulating enzymes such as glucose transporter 1 (GLUT1), GLUT4, hexokinase (HK), pyruvate kinase (PK), and lactate dehydrogenase (LDH). This indicates that HIF-1α can promote tumour cell resistance against radio- and chemo-therapies by induction of glycolysis [98,99,100]. In addition to regulating cellular metabolism, HIF-1α plays a key role in mediating tumour proliferation and angiogenesis [101,102,103]. It also promotes the activation of anti-apoptotic proteins, such as Bcl-2, B-cell lymphoma-extra-large (Bcl-XL), and IAPs, while downregulation of pro-apoptotic BH3 interacting-domain death agonist (Bid) and Bcl-2-associated X protein (Bax) [104]. Moreover, HIF-1α mediates cancer promotion and progression via the activation of histone deacetylases and β-catenin [105]. Thus HIF-1α has multiple ways to activate tumour progression and induce resistance of tumour cells to a range of therapies. By targeting HIF-1α curcumin might sensitise ACHN cells to TRAIL induced apoptosis.

PDK1 is the metabolic key that inhibits pyruvate conversion into acetyl-CoA via phosphorylation and inhibition of the pyruvate dehydrogenase (PDH) enzyme, thus blocking Krebs cycle [106]. It plays an integral role in maintaining glycolysis in cancer cells and contributes to poor prognosis in different cancers including RCC [107,108]. HIF-1α regulates PDK1 mediated glycolytic metabolism in liver-metastatic breast cancer cells, revealing a pivotal role in the progression of liver metastasis [109]. In kidney cancer cells, PDK1 was found to be upregulated, and upon inhibition, the phosphorylation of the oncogenic AKT pathway was inhibited. Therefore, PDK1 promotes cell proliferation and the survival of RCC cells [110]. Furthermore, PDK1 mediates chemoresistance in different cancer cell types by the activation of several oncogenic pathways, namely Polo-like kinase-1 (PLK1)- myelocytomatosis (MYC) Axis, Yes-associated Protein (YAP)/Hippo Pathway, and the serum and glucocorticoid inducible protein kinase (SGK) Axis [111]. Not only does let-7C inhibit aerobic glycolysis by targeting HIF-1α, but it also found to modulate cancer progression and metabolism via inhibiting the lin28/PDK1 pathway [54]. In addition to the direct effect, let-7C can regulate the expression of PDK1 via inhibition of HIF-1α. Let-7C abrogated cancer cell stemness and tumour progression via regulating genes involve in glycolysis as well as those that regulate fatty acid synthesis by targeting lin28/Sterol Regulatory Element-binding Protein-1 (SREBP-1) pathway [80]. We believe that PDK1 inhibition by curcumin shows another important aspect of metabolism regulation, and subsequent chemosensitisation by curcumin. To the best of our knowledge, this is the first study to report the cancer chemosensitisation potential of curcumin on RCC, by the induction of let-7C.

Curcumin is a possible cancer treatment due to its safety, commercial availability, cost effectiveness and bioactivity. However, the poor systemic bioavailability of curcumin represents the main roadblock in using curcumin in clinical settings. Nevertheless, efforts to tackle these issues are currently under investigation, including combinations of curcumin with adjuvants such as piperine, the use of water soluble derivatives, curcumin nanoparticles, as well as liposomal and phospholipid complexes of curcumin [112].

## 4. Materials and Methods

### 4.1. Cell Culture and Treatment

The renal clear cell carcinoma ACHN cells were obtained from the American Type Culture Collection (Manassas, VA, USA). ACHN cells were grown in Minimum Essential Medium (MEM) (Sigma Aldrich, St. Louis, MO, USA) supplemented with 10% fetal bovine serum (FBS) (Gibco, Life Technologies, Carlsbad, CA, USA) and 50 U/mL penicillin and 50 µg/mL streptomycin.

Curcumin (Sigma-Aldrich, St. Louis, MO, USA) in DMSO (Sigma-Aldrich, St. Louis, MO, USA) stock solution was prepared at 50 mM, then diluted with MEM culture medium to 25 uM working concentration. TRAIL (Pepprotech, Rocky Hill, NJ, USA) powder was reconstituted (according to the manufacturer instruction) in sterile distilled water at a concentration of 500 µg/mL. Culture medium was used to prepare the final working concentration (50 ng/mL).

### 4.2. Cell Transfection

ACHN cells were cultured on six-well plates at a density of 2 × 10^5^ cells/well. Next day, cells were transfected with 100 nM let7C miScript miRNA inhibitor (anti-hsa-let-7C-5P) (Qiagen, Lloyd St N, Manchester, UK) or scrambled negative control miRIDIAN (Thermo Fischer Scientific, Waltham, MA, USA) using DharmaFECT (Thermo Fischer Scientific, Waltham, MA, USA) transfecting agent. To prepare the complex, 10 µL of let-7C inhibitor or the scrambled negative control mixed with 5 µL transfecting agent. MEM was used to prepare the complex and to bring the final volume of the complex to 400 µL per a well. Cell transfection was carried out in duplicates. The results represent an average and standard error of three independent experiments.

### 4.3. Long Term Cell Survival Assay

ACHN cells were cultured on 6 well plate at a density of 4 × 10^5^ cells/well. Next day, cells were treated with curcumin 25 µM for 2 h, followed by TRAIL 50 ng/mL for total of 24 h. After incubation, the supernatant was removed, plates were washed then fresh medium was added. After three days, the culture medium was replaced again with fresh medium. Five days post treatment, cells were washed with phosphate buffer saline (PBS), fixed with 4% paraformaldehyde, stained with crystal violet (Sigma Aldrich, St. Louis, MO, USA) and photographed. The absorbance intensity of crystal violet was spectrophotometrically measured at 570 nm using SpectraMax M2 plate reader (Molecular device, Sunnyvale, CA, USA).

### 4.4. Flow Cytometry Analysis

For cell cycle assay, ACHN cells were cultured on six-well plates at a density of 1 × 10^6^ cells/well for 24 h. The cells were then incubated with culture medium containing 0.05% DMSO or 25 µM curcumin for 4 h prior to analysis. Simultaneously, 50% of curcumin-treated cells were further incubated with TRAIL at 50 ng/mL for additional 4, 8, and 20 h (totally 8, 12, 24 h, respectively). Supernatant was then removed, and cells were washed with PBS (Gibco, Life Technologies, Carlsbad, CA, USA), then detached using Trypsin-EDTA (0.25%) (Gibco, Life Technologies, Carlsbad, CA, USA) for 5 min. Trypsin was then neutralised using FBS containing medium, followed by centrifugation of cells. The pellets were re-suspended in PBS and transferred to flow cytometry tubes. The tubes were centrifuged at 1100× *g* revolutions per minute (rpm) for 3 min followed by washing with ice-cold PBS. Cells were then fixed by adding ice-cold 70% ethanol with continuous vortexing. Fixed cells were incubated on ice for one hour followed by washing with PBS, then the pellets were re-suspended with 25 µL RNase (200 µg/mL) (Thermo Fischer Scientific, Waltham, MA, USA) for 5 min followed by adding 500 µL propidium iodide (Sigma-Aldrich, St. Louis, MO, USA) at 50 ng/mL and incubated at 37 °C for 30 min. BD Accuri C6 flow cytometer with the BD Accuri Software (BD Biosciences, San Jose, CA, USA) was employed to perform cell cycle analysis of the samples. BD Accuri C6 calibration was verified with 8 and 6 peak beads as per manufacturer specifications. Propidium iodide was excited with the 488 nm laser, and the emission was collected with a 575/25 and a 670LP filters placed in front of the FL2 and FL3 photomultipliers, respectively. Both signals were compared and the signal of 670LP-FL3 was used as it provided a more sensitive measurement. Data from 30,000 single cells per sample were collected. The cell cycle profile was obtained by using the Multicycle application included in the FCS Express 6 software (DeNovo Software, Glendale, CA, USA). Model 4 was selected for all the files to obtain the statistics.

For MitoTracker™ Red CMXRo (Thermo Fischer Scientific, Waltham, MA, USA) and MitoTracker™ Green FM (Thermo Fischer Scientific, Waltham, MA, USA) assays, cells were treated and collected as described in the cell cycle assay. Cell pellets were resuspended in culture medium containing 50 nM and 100 nM of MitoTracker™ Red CMXRo and MitoTracker™ Green FM dyes, respectively for 45 min. Samples were then acquired using BD Accuri C6 flow cytometry. The MitoTracker Red CMXRos was excited with a 633 nm laser and fluorescence emissions were detected using 675 ± 25 nm nm filter (FL4), while MitoTracker Green was excited with a 48 nm laser and fluorescence emissions were detected using 530/30 nm filter (FL1). Data from 30,000 single cells per sample were collected and re-analysed using DeNovo FCS Express 6 software. Data and gating strategy is shown in Appendix A.


### 4.5. miRNA Extraction, cDNA Synthesis, Pathway Focused miRNA PCR-Array and qRT-PCR

ACHN cells were washed with PBS then total RNA was extracted using miRNeasy mini kit (Qiagen, Lloyd St N, Manchester, UK) according to the manufacturer protocol. The concentration and purity of samples were assessed using a Nanodrop ND-1000 spectrophotometer (Thermo Fischer Scientific, Waltham, MA, USA) according to the manufacturer instructions. miRNA integrity was checked using a Bioanalyser (Agilent, Santa Clara, CA, USA) according to the manufacturer protocol. A minimum amount of 250 ng of total RNA was employed to make cDNA using miScript II RT kit (Qiagen, Lloyd St N, Manchester, UK) according to the manufacturer protocol. For miRNA PCR-array, standard apoptosis pathway focused 384-well PCR arrays (Qiagen, Lloyd St N, Manchester, UK) were employed according to the manufacturer instructions. Data were uploaded and analysed using miScript miRNA PCR-array data analysis provided by Qiagen. For qRT-PCR, miScript SYBR Green PCR Kit (Qiagen, Lloyd St N, Manchester, UK) was used for PCR reaction according to the manufacturer protocol. Briefly, 9 µL reaction solution containing 5 µL of 2× QuantiTect SYBR Green PCR Master Mix, 1 µL 10× miScript Universal Primer, 1 µL RT2 qPCR let-7C primer or RUN6B (Qiagen, Lloyd St N, Manchester, UK) loading control, and 2 µL water. The reaction mixture was loaded in an optical 384 well plate in duplicates followed by loading 1 µL of cDNA samples. The reaction was set according to the QuantiFast SYBR^®^ Green RT-PCR Kit manufacturer instructions. The abundance of let-7C was normalised against RUN6B loading control. The 2^−ΔΔCt^ method was employed to analyse the results [38].

### 4.6. Computational Target Prediction In Silico Analysis

Three major target prediction tools including Target scan V7.2 (Whitehead Institute for Biomedical Research, Cambridge, MA, USA) [113] PicTar (Center for Comparative Functional Genomics and the Max Delbruck Centrum, Berlin, Germany) [114] and miRDB (Washington University, St. Louis, MO, USA) [115] were employed to predict potential targets of the deregulated miRNAs.

### 4.7. Western Blot Analysis

Western blot analysis was undertaken according to a previously described protocol [116]. Cells were lysed with radio immunoprecipitation assay (RIPA) analysis buffer (Sigma-Aldrich, St. Louis, MO, USA) and total protein concentrations were determined using a bicinchoninic acid assay (BCA assay) protein assay kit (Pierce, Rockford, IL, USA) according to the manufacturer’s protocol. Samples were subjected to sodium dodecyl-sulfate polyacrylamide gel electrophoresis (SDS-PAGE) electrophoresis, then transferred to a 0.2 µm pore size nitrocellulose membrane Whatman Protran^®^ (Thermo Fisher Scientific, Waltham, MA, USA) using a SemiPhor semi-dry transfer system (G. E. Healthcare, Chicago, IL, USA). Membranes were incubated for one hour at room temperature with 5% non-fat milk (Sigma-Aldrich, St. Louis, MO, USA) or 5% bovine serum albumin (BSA) (Sigma-Aldrich, St. Louis, MO, USA). Membranes were then incubated overnight at 4 °C with a primary antibody against (β-catenin, CDK1, CDK2, CDK4, CDK6, cyclin D1, cyclin B1, HIF-1α, PDK) Santa Cruz, Dallas, Texas, USA; all 1:1000 dilution except HIF-1 α dilution 1:500). GAPDH (Cell Signalling Technology Inc., Danvers, MA, USA) was employed at 1:1000 dilution as a loading control. Following this, blots were washed and incubated with an appropriate species horseradish peroxidase (HRP)-linked secondary antibody at a dilution of 1:2000 dilution (Cell Signalling Technology Inc., Danvers, MA, USA) for 1 h at room temperature. SuperSignal™ West Pico Chemiluminescent Substrate (Thermo Scientific, Rockford, IL, USA) was used to detect signals via chemiluminescent detection method. Image J software 1.50i (National Institutes of Health, Bethesda, MD, USA) (http://imagej.nih.gov/ij, accessed on 23 February 2022) was harnessed to perform the densitometric analysis.

### 4.8. Statistical Analysis

All data were analysed using GraphPad prism 5.0 (Graph Pad software, San Diego, CA, USA). Where appropriate, ANOVA (one and two way) and paired t-test were used to analyse results and calculate the level of significance. Results were expressed as the mean ± standard error of the mean (SEM). A probability of 0.05 or less was considered as statistically significant.

## 5. Conclusions

Curcumin is a promising chemopreventive, anticancer, and chemosensitising agent. It potentially deregulates several signaling pathways in renal cancer cells including miRNAs. Curcumin showed significant inhibition of cell cycle progression at G1, S, and G2/M phases. It also promoted the activation of mitochondrial respiration while inhibiting the glycolytic pathway. Further, curcumin upregulated several miRNAs that regulate cell cycle, metabolism and cell death including let-7C. However, more in vivo as well as preclinical studies are warranted to validate the let-7C induction by curcumin. Further, more experiments are needed to explore the mechanism of let-7C induction by curcumin in RCC and therapeutic potential. There are some limitations associated with this study. Firstly, it would be of great importance to investigate the effects of curcumin and let-7C antagomir treatments on patient-derived primary renal carcinoma cells in parallel with the ACHN cell line. Further, targeting of cells with the miRNA antagomir to confirm the involvement of let-7C in the regulation of cell cycle and metabolism is another important consideration. Therefore, we plan to investigate let-7C mimics in a large in vitro, in vivo and ex vivo study. We believe that this study has provided some basis for further mechanistic studies to explore other tumour suppressive mechanisms of let-7C in RCC. Furthermore, it encourages future initiatives to investigate the effects of other miRNAs such as miRNA 30d, miRNA 210 and miRNA 542 in RCC. Finally, this study raises the possibility to gain further insight into the molecular mechanisms of interrupting cell cycle and mitochondrial metabolism by Let-7C in RCC.

## Figures and Tables

**Figure 1 ijms-23-09569-f001:**
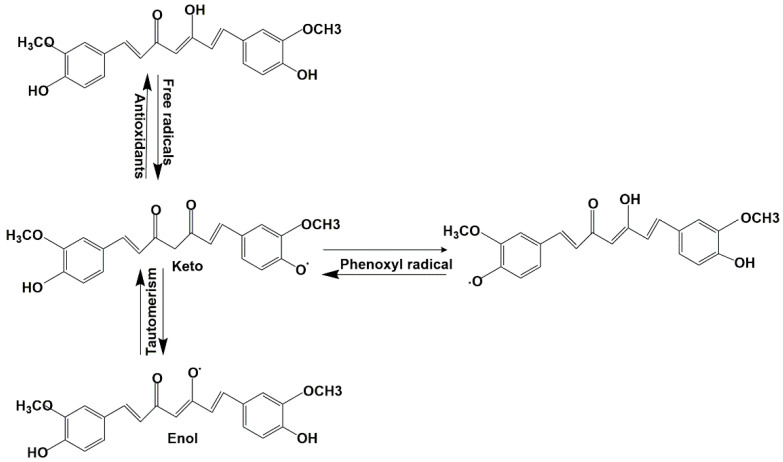
Structure and antioxidant mechanism of curcumin. The structures were drawn using CD Chemdraw v 15.1.0.144 (PerkinElmer, Waltham, MA, USA).

**Figure 2 ijms-23-09569-f002:**
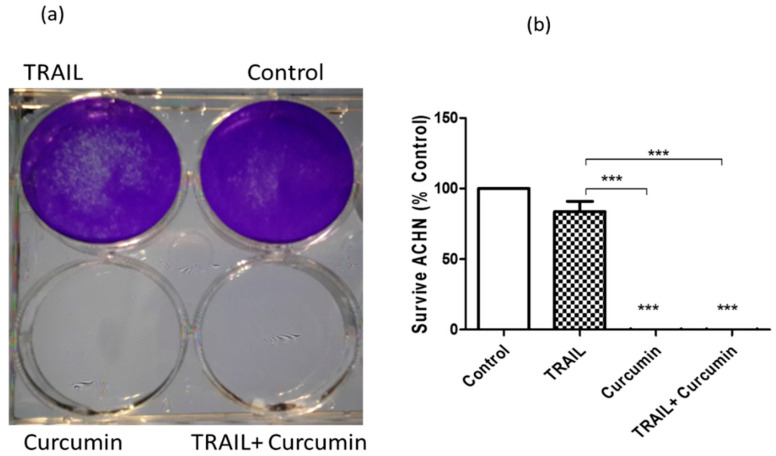
Effect of curcumin on the long-term survival of cancerous ACHN cells. Cells were treated either with 25 µM curcumin, 25 ng/mL TRAIL or a combination of both, then stained five days post treatment with 0.1% crystal violet. (**a**) Photographic image of the plate: clear or colorless wells indicates no cells while purple stain refers to the presence of cellular growth. (**b**) Absorption intensity of crystal violet was measured spectrophotometrically and results are expressed as relative survival growth (% of control) ± standard error of the mean (SEM) of five independent experiments. *** indicate statistically significant difference at *p* < 0.001 compared to DMSO or TRAIL-treated cells. Data were analysed using ONE-WAY analysis of variance (ANOVA).

**Figure 3 ijms-23-09569-f003:**
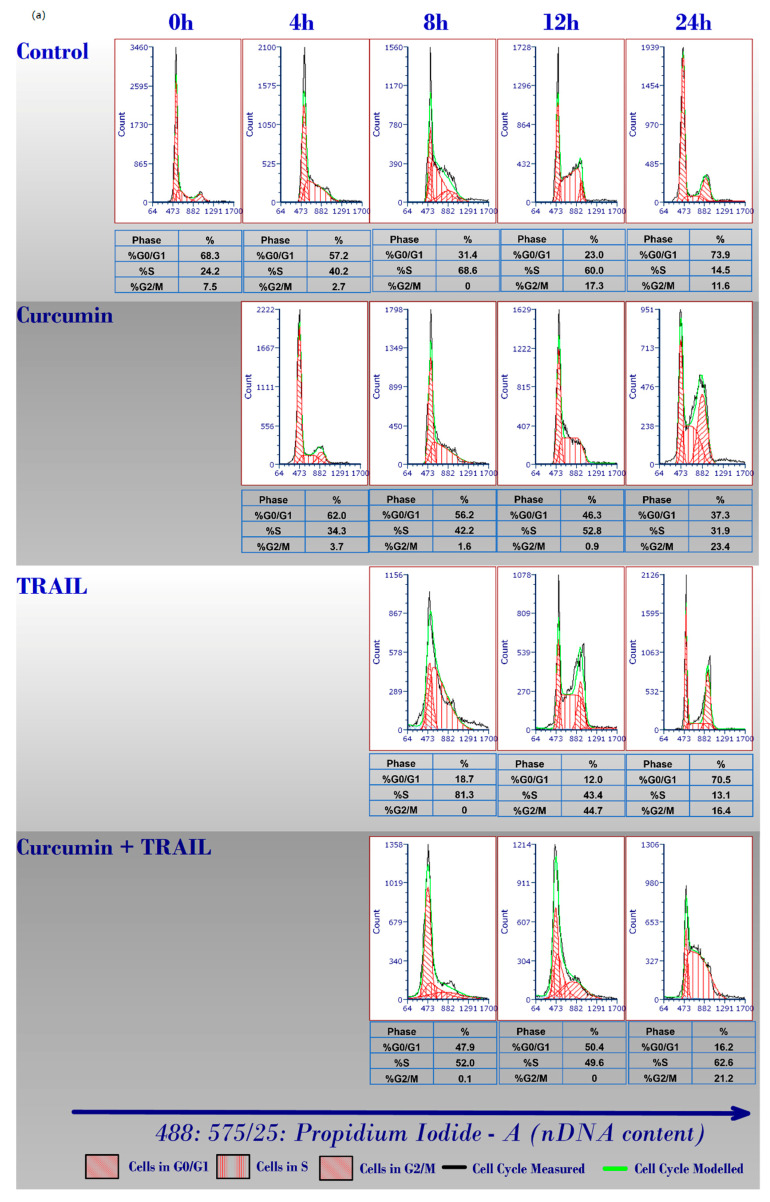
Effects of curcumin, TRAIL, curcumin+ TRAIL combination on ACHN cell cycle. Cell cycle was examined by flow cytometry with a solid-state blue laser (488 nm) and fluorescence emissions were detected using 575/25 nm filter (FL2). Cells were fixed 70% ethanol and nuclear DNA (nDNA) was labelled with Propidium Iodide (PI). (**a**) Representative histograms show the percentage of cells residing at each cell cycle (G0/G1, S, and G2/M) phase following the exposure to DMSO, TRAIL, curcumin and curcumin+ TRAIL for different times. At the beginning, cell cycle profile was assessed before adding any treatment (a-0 h). Curcumin was then added for 4 h, and cell cycle profile was measured and compared with the corresponding untreated control (a-4 h). After 4 h, TRAIL was added and incubated for a further 4 h (a-8 h), 8 h (a-12 h) or 20 h (a-24 h). Data were analysed using TWO-WAY ANOVA. (**b**) Statistical analysis of cell cycle histograms. Each histogram shows the percentage of cells residing at each cell cycle phase following exposure to the indicated treatments. Cell populations accumulated at each cell cycle (G0/G1, S, and G2M) phase were measured, analysed, and compared. Data are expressed as mean ± SEM of three independent experiments. * *p* < 0.05, ** *p* < 0.01, *** *p* < 0.001 compared to the vehicle control or TRAIL treated cells. Different bar patterns were used to indicate different treatments and are indicated in the figure.

**Figure 4 ijms-23-09569-f004:**
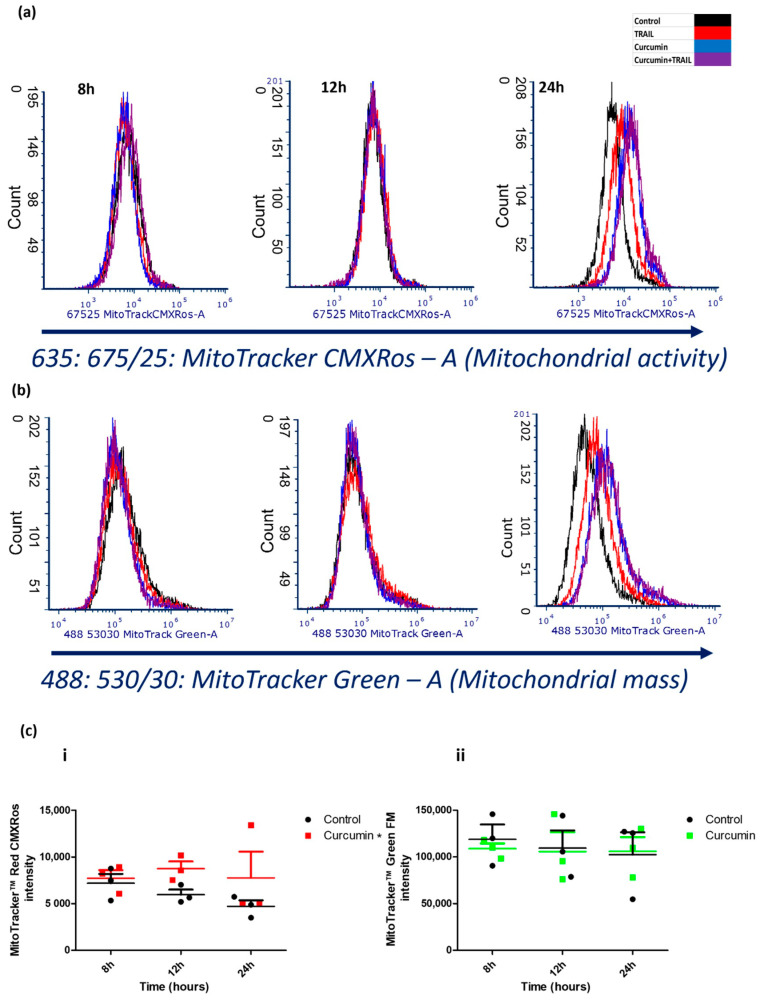
Effect of curcumin on the mitochondrial mass and activity in ACHN cells. MitoTracker Red CMXRos and MitoTracker Green were employed to measure mitochondrial activity and mass using flow cytometry. MitoTracker Red CMXRos was excited with a solid-state red laser (635 nm) and fluorescence emissions were detected using 675/25 nm filter (FL4) while MitoTracker Green was excited with a solid-state blue laser (488 nm) and fluorescence emissions were detected using 530/30 nm filter (FL1). (**a**,**b**) representative single-parameter overlay histograms after 8, 12, and 24 h, respectively. (**c**) Column scatter chart shows the effect of curcumin on the mitochondrial activity (**i**) and mitochondrial mass (**ii**). The data are presented as mean of fluorescence intensity ± SEM of three independent experiments. * *p* < 0.05 compared to untreated control. Data were analysed using TWO-WAY ANOVA.

**Figure 5 ijms-23-09569-f005:**
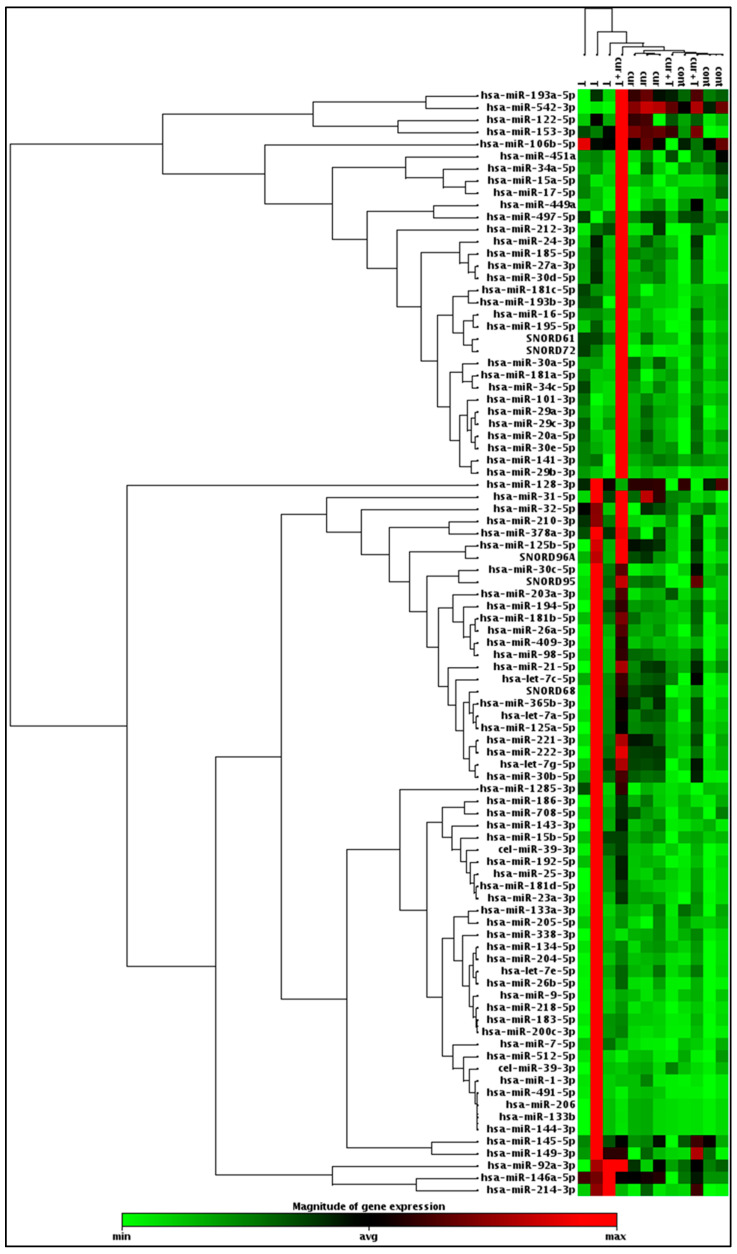
Hierarchical clustering analysis of miRNA microarray data. The heat map diagram shows the results of the two-way hierarchical clustering analysis of 84 apoptosis-regulating miRNAs after targeting ACHN cells with curcumin (cur), TRAIL (t) and TRAIL and curcumin (cur + T). The color scale illustrates the relative expression level of miRNAs ranging from high (red) to low (green) relative expression.

**Figure 6 ijms-23-09569-f006:**
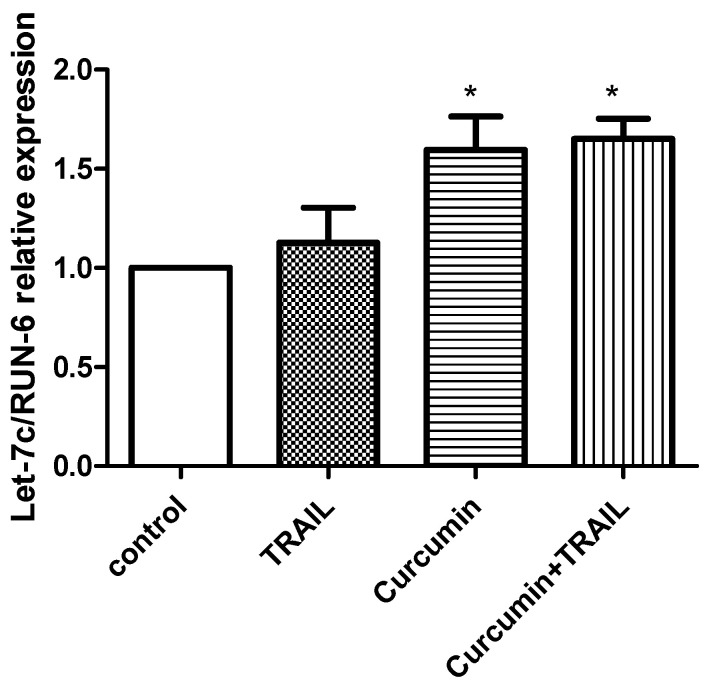
Validation of let-7C upregulation by curcumin or curcumin+ TRAIL combination. Quantitative RT-PCR was employed to validate the upregulation of let-7C in cancerous ACHN cells treated with curcumin or curcumin+ TRAIL. Data are expressed as mean ± SEM of three independent experiments. * *p* < 0.05 vs. vehicle treated control. Data were analysed using Paired-*t*-test and ONE-WAY ANOVA.

**Figure 7 ijms-23-09569-f007:**
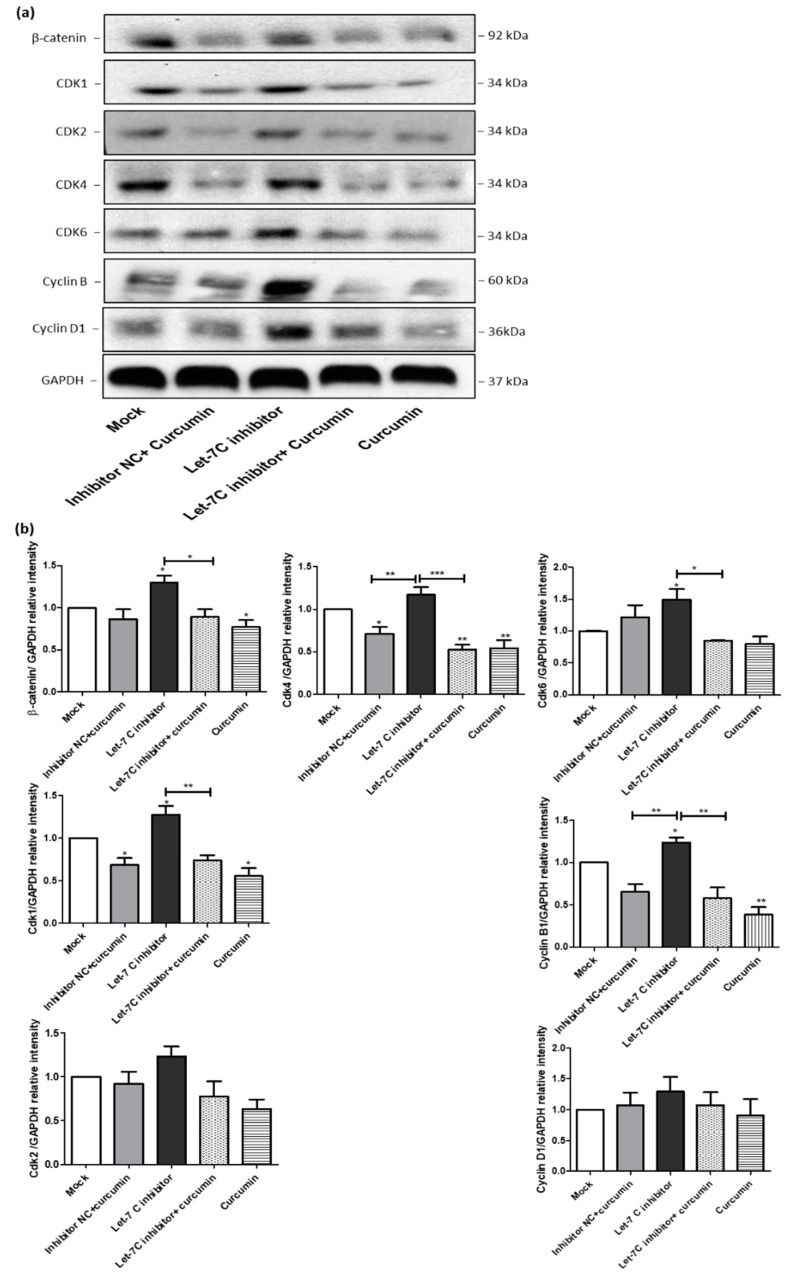
Effect of let7C inhibition on cell cycle regulating proteins. (**a**) A representative western blot is shown from three independent let-7C transfection experiments where expression of β-catenin, CDK1, CDK2, CDK4, CDK6, cyclin B and cyclin D1 was examined. (**b**) Band intensity of the indicated proteins were quantified densitometrically, normalised to glyceraldehyde-3-phosphate dehydrogenase (GAPDH) level and expressed as the mean relative intensity ± SEM (*n* = 4). * *p* < 0.05, ** *p* < 0.01, and *** *p* < 0.001. ONE-WAY ANOVA was used to analyse the data.

**Figure 8 ijms-23-09569-f008:**
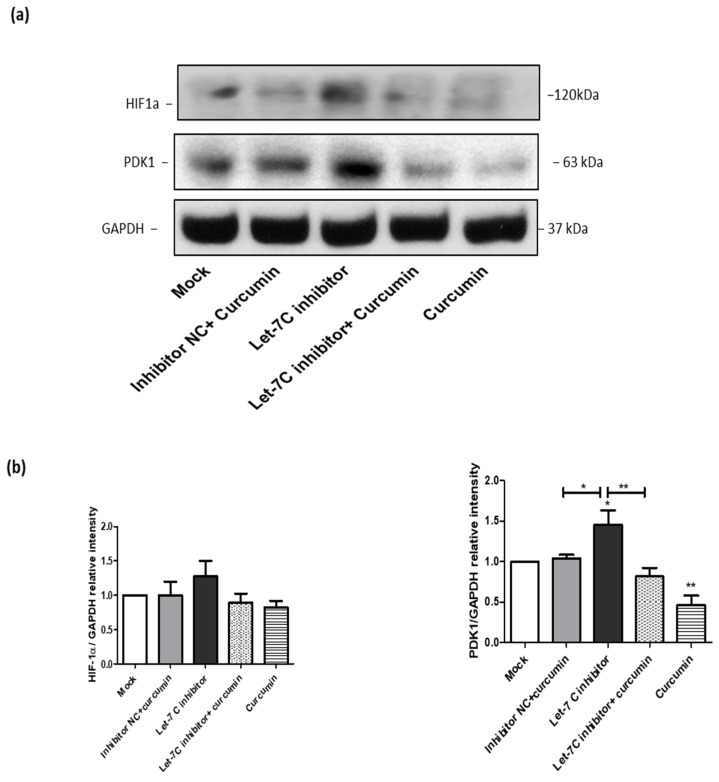
The effect of let-7C inhibition on glycolysis regulating proteins. (**a**) A representative western blot is shown from three independent let-7C transfection experiments, where expression of HIF-1α and PDK1 was examined. (**b**) Band intensity of the indicated western blots were quantified densitometrically, normalized to GAPDH level and expressed as the mean relative intensity ±SEM (*n* = 3). * *p*< 0.05 and ** *p* < 0.01. ONE-WAY ANOVA was used to analyse the data.

## Data Availability

Not applicable.

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
