# Peer review of "Curcumin Sensitises Cancerous Kidney Cells to TRAIL Induced Apoptosis via Let-7C Mediated Deregulation of Cell Cycle Proteins and Cellular Metabolism"

_ijms, 2022, doi:10.3390/ijms23179569_

Round 1
Reviewer 1 Report
Title: Curcumin sensitised cancerous kidney cells to TRAIL induced apoptosis via let-7C mediated deregulation of cell cycle proteins and cellular metabolism
Authors: Ismael Obaidi, Tara McMorrow, Alfonso Blanco Fernández
Comments: The authors have reported in previous studies that curcumin sensitizes renal carcinoma ACHN to the proapoptotic effects of TRAIL via activation of ROS/DR4 and ROS/JNK/CHOP signaling pathways. In this study, they investigated the involvement of miRNAs in the sensitization of ACHN cells by curcumin and examined its effects on cell cycle and metabolism.
Key Points:
- Page 1 line 39-42: a brief explanation of the mechanism of action would be nice here and why other tissues are not affected.
- Page 2 lines 52-54: some more background information on curcumin would be good (origin/manufacture, chemical structure, mode of action, etc.)
- Page 2 line 58-59: here is a very abrupt jump from curcumin to microRNAs, there needs to be a transition between the two topics
- Page 2 line 82-85: Here I miss the explanation of an overall goal of the study
- Page 2 lines 87-89: in my opinion redundant
- Page 3 Figure 1: Here I miss a more detailed explanation in the text or legend: what can be seen; what does the coloring/no coloring mean? What does it mean?
- Page 3 line 108/109: There should be a brief explanation of what was done/introduction here first, instead of listing a result right in the first sentence without an introduction.
- Page 4/5 Figure 2A/B: here I miss more detailed descriptions of the figures as they are very complex.
- Page 6 lines 137-138: what does this result mean? What does it point to? What does it indicate?
- Page 6 Figure 3A: the labeling of the x-axis is not really meaningful, it needs to be made clear what the MitoTracker is measuring in terms of mitochondrial activity
- Page 8 Figure 4: should be briefly explained in the body text.
- Page 9 line 179: chapter 2.6. is very short here
- Page 9 line 186-195: should be briefly explained here what methods were used (Figure 6A shows a Western blot, but is not mentioned as such in the text)
- Same page 11 Figure 7A
- Page 12 discussion: formatting!!!
- Page 12 Discussion: Introduction is missing: what was the actual background of the study? Why was what done and how?
- Page 12 line 256: what does this result mean? Interpretation? Discussion?
- Page 13 line 288-289: jumpy transition; transition missing.
- Page 13 line 301 and line 302 + 305: different formatting of references than in the rest of the paper.
- Conclusion: what are the limitations of the study or its significance? What was accomplished and with what perspective?
Author Response
General Feedback on actions to the reviewer.
Dear reviewer,
Thank you very much for your feedback and comments on the submitted manuscript. We apologise for the errors that came through to you and can assure you that they have all been corrected now. We have carried out a substantial revision of the manuscript. We have rearranged some text which was highlighted in your comments and have inserted some additional text and commentary in the introduction whichadds clarity to the manuscript.
Specific comments
Reviewer# 1
- Page 1 line 39-42: a brief explanation of the mechanism of action would be nice here and why other tissues are not affected.
A detailed explanation of the mechanism of action of TRAIL including “why other tissues are not affected” has been added to the manuscript as requested
- Page 2 lines 52-54: some more background information on curcumin would be good (origin/manufacture, chemical structure, mode of action, etc.)
Information on curcumin, including its’ chemical structure has been added to the manuscript as requested
- Page 2 line 58-59: here is a very abrupt jump from curcumin to microRNAs, there needs to be a transition between the two topics
Extra sentences have been included in the text to support the transition between the two topic: curcumin and miRNA
- Page 2 line 82-85: Here I miss the explanation of an overall goal of the study
We have added a paragraph to explain the overall goal of the study
- Page 2 lines 87-89: in my opinion redundant
We apologies for this redundant section , this has been removed now
- Page 3 Figure 1: Here I miss a more detailed explanation in the text or legend: what can be seen; what does the coloring/no coloring mean? What does it mean?
Sincere apologies for the ambiguity, we have added more detail in the text and legends to make them clearer
- Page 3 line 108/109: There should be a brief explanation of what was done/introduction here first, instead of listing a result right in the first sentence without an introduction.
We acknowledge this point. We have added an introductory paragraph
- Page 4/5 Figure 2A/B: here I miss more detailed descriptions of the figures as they are very complex.
The figure label was modified, and some details have been added to the legend to make it clearer
- Page 6 lines 137-138: what does this result mean? What does it point to? What does it indicate?
An explanation has been added to the text to answer the reviewer’s important queries
- Page 6 Figure 3A: the labeling of the x-axis is not really meaningful, it needs to be made clear what the MitoTracker is measuring in terms of mitochondrial activity
The authors agree with this comment. We have kept the information about laser, filter and dye used as recommended by ISAC (International Society for the Advancement of Cytometry), but we have simplified to one single X-axis. We have added as a general title what is measured with these markers. We hope this is sufficient, however if the reviewers require further clarification, we can include more information.
- Page 8 Figure 4: should be briefly explained in the body text.
The figure has been explained in the text
- Page 9 line 179: chapter 2.6. is very short here
More details have been added to this section
- Page 9 line 186-195: should be briefly explained here what methods were used (Figure 6A shows a Western blot, but is not mentioned as such in the text)
Methods used, including western blotting, have been mentioned and explained
- Same page 11 Figure 7A
Western blot has been briefly mentioned and explained
- Page 12 discussion: formatting!!!
Reformatting of discussion has been accomplished
- Page 12 Discussion: Introduction is missing: what was the actual background of the study? Why was what done and how?
Multiple paragraphs have been added which we hope answer all the reviewer’s valuable comments
- Page 12 line 256: what does this result mean? Interpretation? Discussion?
An explanation has been added to answer the reviewer’s insightful comment
- Page 13 line 288-289: jumpy transition; transition missing.
Two separate paragraphs have been added to make a transition between points
- Page 13 line 301 and line 302 + 305: different formatting of references than in the rest of the paper.
Sincere apologies for these references. They have been modified now
- Conclusion: what are the limitations of the study or its significance? What was accomplished and with what perspective?
We have added details to answer these important questions
Reviewer 2 Report
In this study authors evaluated the role of curcumin in sensitising TRAIL-resistant kidney cancerous ACHN cells to TRAIL and found that curcumin deregulated the expression of apoptosis-regulating miRNA, in particular let-7C. Transfecting ACHN cells with Let-7C antagomir significantly overexpressed the expression of several cell cycle and glycolysis regulating proteins.
Manuscript is interesting but it presents some flaws that must be resolved. In particular:
Lines 52-58: In the introduction authors should highlight the pleiotropic role of curcumin since it does not only have anticancer activity but it also plays a key role in insulin sensitivity, as an anti-obesity agent and has beneficial effects on pregnancy outcome (PMID: 34981472, 33477354,31781039). This is an important information to point out since it can further highlight the importance of the results found by the authors.
Lines 87-89: Please remove because are part of the template file of the journal
Figure 1: in the previous study published by the authors, they reported that in 25 uM Curcumin treatment the 65% of cells survived and in the TRAIL+Curcumin treatment survived cells dropped to 13.9% (PMID: 32370057). Here the treatment with curcumin at same molarity killed all cells thus there was nothing to sensitize to TRAIL since all cells were death already before TRAIL treatment. It is clear that something went wrong. Please clarify.
Figure 6: Authors must show an apoptotic marker (e.g Cleaved Caspase 3) because Curcumin alone showed a significant cell death in cell viability assay and Flow cytometry analysis. Moreover, when asterisks are shown, author must use bars to show what has been compared.
Authors are suggested to expand all acronyms when they first appear in the text and maintain acronyms only throughout the text.
Author Response
General Feedback on actions to the reviewer.
Dear reviewer,
Thank you very much for your feedback and comments on the submitted manuscript. We apologise for the errors that came through to you and can assure you that they have all been corrected now. We have carried out a substantial revision of the manuscript. We have rearranged some text which was highlighted in your comments and have inserted some additional text and commentary in the introduction whichadds clarity to the manuscript.
Specific comments
Reviewer# 2
Figure 1: in the previous study published by the authors, they reported that in 25 uM Curcumin treatment the 65% of cells survived and in the TRAIL+Curcumin treatment survived cells dropped to 13.9% (PMID: 32370057). Here the treatment with curcumin at same molarity killed all cells thus there was nothing to sensitize to TRAIL since all cells were death already before TRAIL treatment. It is clear that something went wrong. Please clarify.
This is an excellent question. From our previous study (PMID: 32370057), as the total time of the viability assay (from seeding cells to reading the plates) was 48 hours, we were using the highest possible cell density, which was 7.5x10^5 cells/ml, to ensure that cells were 90-100% confluent before adding treatments so that cell death due to low seeding density was avoided. However, in the current study, we were following protocols (cited in the manuscript) from published articles (PMID:19089423 and 25719250) for seeding and treating cells for clonal outgrowth or long-term cell viability experiment. Because the experiment is set for seven days, cells should be seeded at a lower cell density which is 2.5x10^4 cells/well for 24 well plate or 1-1.25x10^5 cells/well in 6-well plate format, instead we seeded cells at approximately four times higher (4x10^5 cells/well) than the seeding density mentioned in these articles. However, cell death for the entire population was still observed suggesting that curcumin has a powerful cell density-dependent cytotoxic effect in addition to its’ chemosensitising potential. Nevertheless, the capability to distinguish between anticancer and chemosensitising activities of curcumin is an important limitation of this assay.
Figure 6: Authors must show an apoptotic marker (e.g Cleaved Caspase 3) because Curcumin alone showed a significant cell death in cell viability assay and Flow cytometry analysis. Moreover, when asterisks are shown, author must use bars to show what has been compared.
We acknowledge the reviewer comment. In our previous study (PMID: 32370057), we showed the cleaved nuclear apoptotic marker PARP by curcumin+ TRAIL combination but not by either curcumin or TRAIL. In the same study, curcumin induced a transient and reversible apoptotic event according to the flow cytometric analysis of apoptosis. In contrast, cell cycle analysis of the present study revealed that curcumin is the key player in arresting cell cycle progression which is in agreement with other studies (PMID: 31752145, PMID: 33649826, PMID: 32277661, PMID: 29673545). Further, flow cytometric analysis of mitochondrial activity in the present study clearly showed that curcumin had no direct effect on mitochondrial size, instead, the combinatorial treatment of curcumin+ TRAIL significantly increased both the size and activity of mitochondria
In relation to the asterisks, to avoid complexity, if the comparison with the control, then the asterisks were placed on the bars, otherwise we drew a line between each two treatments and the asterisks were placed on top of the line. We have used this same way of expressing significance in our previous publications (PMID: 32370057 and PMID: 29278208). Further, this way of indicating asterisks can be found in other publications (PMID: 18509548, PMID: 35094294, PMID: 32229665). However, if preferred, we can change all the figures to satisfy the reviewer.
Authors are suggested to expand all acronyms when they first appear in the text and maintain acronyms only throughout the text.
All acronyms have been expanded when they appear first in the text
Round 2
Reviewer 1 Report
The authors have satisfactorily addressed the concerns raised in the original version. The revised version is significantly improved. No further concerns.
Reviewer 2 Report
the manuscript has been significantly improved and can be accepted in the present form.